# Mitigating the Impact of Labeling Errors on Training via Rockafellian Relaxation

## Abstract

Labeling errors in datasets are common, if not systematic, in practice. They naturally arise in a variety of contexts—human labeling, noisy labeling, and weak labeling (i.e., image classification), for example. This presents a persistent and pervasive stress on machine learning practice. In particular, neural network (NN) architectures can withstand minor amounts of dataset imperfection with traditional countermeasures such as regularization, data augmentation, and batch normalization. However, major dataset imperfections often prove insurmountable. We propose and study the implementation of Rockafellian Relaxation (RR), a new loss reweighting, architecture-independent methodology, for neural network training. Experiments indicate RR can enhance standard neural network methods to achieve robust performance across classification tasks in computer vision and natural language processing (sentiment analysis). We find that RR can mitigate the effects of dataset corruption due to both (heavy) labeling error and/or adversarial perturbation, demonstrating effectiveness across a variety of data domains and machine learning tasks.

## 1 Introduction

Labeling errors are systematic in practice, stemming from various sources. For example, the reliability of human-generated labels can be negatively impacted by incomplete information, or the subjectivity of the labeling task - as is commonly seen in medical contexts, in which experts can often disagree on matters such as the location of electrocardiogram signal boundaries [8], prostate tumor region delineation, and tumor grading [20]. As well, labeling systems, such as Mechanical Turk[1] often find expert labelers being replaced with unreliable non-experts [27]. For all these reasons, it would be advisable for any practitioner to operate under the assumption that their dataset is corrupted with labeling errors, and possibly to a large degree.

In this paper, we propose a loss-reweighting methodology for the task of training a classifier on data having higher levels of labeling errors. We show that our method relates to optimistic and robust distributional optimization formulations aimed at addressing adversarial training (AT). These findings underscore our numerical experiments on NNs that suggest this method of training can provide test performance robust to high levels of labeling error, and to some extent, feature perturbation. Overall, we tackle the prevalent challenges of label corruption and class imbalance in training datasets, which are critical obstacles for deploying robust machine learning models. Our proposed approach implements Rockafellian Relaxations [23] to address corrupted labels and automatically manage class imbalances without the need for clean validation sets or sophisticated hyper-parameters - common constraints of current methodologies. This distinct capability represents our key contribution, making our approach more practical for handling large industrial datasets.

---

[1]http://mturk.com

Submitted to 38th Conference on Neural Information Processing Systems (NeurIPS 2024). Do not distribute.

We proceed to discuss related works in section 2, and our specific contributions to the literature. In section 3 we discuss our methodology in detail and provide some theoretical justifications that motivate the effectiveness of our methodology. The datasets and NN model architectures upon which our experimental results are based are discussed in sections 4 and 5, respectively. We then conclude with numerical experiments and results in section 6.

## 2  Related Work

Corrupted datasets are of concern, as they potentially pose severe threats to classification performance of numerous machine-learning approaches [36], including, most notably, NNs [15, 33]. Naturally, there have been numerous efforts to mitigate this effect [28, 8]. These efforts can be categorized into robust architectures, robust regularization, robust loss function, *loss adjustment*, and sample selection [28]. Robust architecture methods focus on developing custom NN layers and dedicated NN architectures. This differs from our approach, which is architecture agnostic and could potentially "wrap around" these methods. While robust regularization methods like data augmentation [26], weight decay [16], dropout [29], and batch normalization [14] can help to bolster performance, they generally do so under lower levels of dataset corruption. Our approach, on the other hand, is capable of handling high levels of corruption, and can seamlessly incorporate methods such as these. In label corruption settings, it has been shown that loss functions, such as robust mean absolute error (MAE) [10] and generalized cross entropy (GCE) [35] are more robust than categorical cross entropy (CCE). Again, our method is not dependent on a particular loss function, and it is possible that arbitrary loss functions, including robust MAE and GCE, can be swapped into our methodology with ease. Our approach resembles the loss adjustment methods most closely, where the overall loss is adjusted based on a (re)weighting scheme applied to training examples.

In *loss adjustment* methods, individual training example losses are typically adjusted multiple times throughout the training process prior to NN updates. These methods can be further grouped into loss correction, loss reweighting, label refurbishment, and meta-learning [28]. Our approach most closely resembles the *loss reweighting* methods. Under this scheme each training example is assigned a unique weight, where smaller weights are assigned to examples that have likely been corrupted. This reduces the influence of corrupted examples. A training example can be completely removed if its corresponding weight becomes zero. Indeed, a number of loss reweighting methods are similar to our approach. For example, Ren et al., [22] learn sample weights through the use of a noise-free validation set. Chang et al. [5] assign sample weights based on prediction variances, and Zhang et al. [34] examine the structural relationship among labels to assign sample weights. However, we view the need for a clean dataset, or at least one with sufficient class balance, by these methods as a shortcoming, and our method, in contrast, makes no assumption on the availability of such a dataset.

Satoshi et al. [12] propose a two-phased approach to noise cleaning. The first phase trains a standard neural network to determine the top-$m$ most influential training instances that influence the decision boundary; these are subsequently removed from the training set to create a cleaner dataset. In the second phase, the neural network is retrained using the cleansed training set. Their method demonstrates superior validation accuracy for various values of $m$ on MNIST and CIFAR-10. Although impressive, their method does not address the fact that most industrial datasets have a reasonably large amount of label corruption [28] which, upon complete cleansing, could also remove informative examples that lie close to the decision boundary. Additionally, the value of $m$ is an additional hyper-parameter that could require significant tuning on different datasets and sources.

Mengye et al. [22] propose dealing with label noise and class imbalance by learning exemplar weights automatically. They propose doing so in the following steps: a) Create a pristine noise-free validation set. b) Initially train on a large, noisy training dataset, compute the training loss on the training set, train on the clean validation set, and compute the training loss on the validation set. c) Finally, compute the exemplar weights that temper the training loss computed in step two with validation loss. This approach is algorithmically the most similar to ours, with some key differences. The major difference is that it treats noise and class imbalance similarly. Our approach deals with noisy labels explicitly and can cope with almost any amount of class imbalance automatically, as tested in our experiments with the open-source Hate-Speech dataset, where we experimented with different prevalence levels of Hate-Speech text. The biggest drawback of the method proposed by Mengye et al. is that it requires a clean validation set, which in practice is almost impossible

to obtain; if it were possible, it would not be very prohibitive to clean the entire dataset. Noise, typically, is an artifact of the generative distribution which cannot be cherry-picked as easily in practice. Our approach does not require a clean dataset to be operational or effective.

# 3 Methodology

## 3.1 Mislabeling

Let $\mathcal{X}$ denote a *feature* space, with $\mathcal{Y}$ a corresponding *label* space. Then $\mathcal{Z} := \mathcal{X} \times \mathcal{Y}$ will be a collection of feature-label pairs, with an unknown probability distribution $D$. Throughout the forthcoming discussions, $\{(x_i, y_i)\}_{i=1}^N$ will denote a sample of $N$ feature-label pairs, for which some pairs will have a mislabeling. More precisely, we begin with a collection $(x_i, \tilde{y}_i)$ drawn i.i.d. from $D$, but there is some unknown set $C \subsetneq \{1, \ldots, N\}$ denoting (corrupted) indices for which $y_i = \tilde{y}_i$ if and only if $i \notin C$. For those $i \in C$, $y_i$ is some incorrect label, selected uniformly at random, following the Noise Completely at Random (NCAR) model [8] also known as *uniform label noise*.

## 3.2 Rockafellian Relaxation Method (RRM)

We adopt the empirical risk minimization (ERM) [31] problem formulation:

$$\min_{\theta} \frac{1}{N} \sum_{i=1}^N J(\theta; x_i, y_i) + r(\theta) \tag{1}$$

as a baseline against which our method is measured. Given an NN architecture with (learned) parameter setting $\theta$ that takes as input any feature $x$ and outputs a prediction $\hat{y}$, $J(\theta; x, y)$ is the loss with which we evaluate the prediction $\hat{y}$ with respect to $y$. Finally, $r(\theta)$ denotes a regularization term.

In ERM it is common practice to assign each training observation $i$ a probability $p_i = 1/N$. However, when given a corrupted dataset, we may desire to remove those samples that are affected; in other words, if $C \subsetneq \{1, ..., N\}$ is the set of corrupted training observations, then we would desire to set the probabilities in the following alternative way:

$$p = (p_1, ..., p_N) \text{ with } p_i = \begin{cases} 0, & \text{if } i \in C \\ \frac{1}{N - |C|}, & \text{if } i \in \{1, ..., N\} \setminus C, \end{cases} \tag{2}$$

where $|C|$ is the cardinality of the unknown set $C$. In this work, we provide a procedure - the *Rockafellian Relaxation Method* (RRM) - with the intention of aligning the $p_i$ values closer to the desired (but unknown) $p$ of (2) in self-guided, automated fashion. It does so by adopting the Rockafellian Relaxation approach of [23]. More precisely, we consider the problem

$$\min_{\theta} \left[ v(\theta) := \min_{u \in U} \sum_{i=1}^N (\frac{1}{N} + u_i) \cdot J(\theta; x_i, y_i) + \gamma \|u\|_1 \right], \tag{3}$$

where $U := \{u \in \mathbb{R}^N : \sum_{i=1}^N u_i = 0, \frac{1}{N} + u_i \geq 0 \ \forall i = 1, \ldots, N\}$, and some $\gamma > 0$.

We proceed to comment on this problem that is nonconvex in general, before providing an algorithm.

## 3.3 Analysis and Interpretation of Rockafellian Relaxation

Although problem (3) is nonconvex in general, the computation of $v(\theta)$ for any fixed $\theta$ amounts to a linear program. The following result characterizes the complete set of solutions to this linear program, and in doing so, provides an interpretation of the role that $\gamma$ plays in the loss-reweighting action of RRM.

**Theorem 3.1.** *Let $\gamma > 0$ and $c = (c_1, \ldots, c_N) \in \mathbb{R}^N$, with $c_{min} := \min_i c_i$, and $c_{max} := \max_i c_i$. Write $I_{min} := \{i : c_i = c_{min}\}$, $I_{big} := \{i : c_i = c_{min} + 2\gamma\}$, and for any $S_1 \subseteq I_{min}, S_2 \subseteq I_{big}$,*

126 *define the polytope* $U^*_{S_1,S_2} := \left\{ u^* \in U : \begin{array}{l} u^*_i \geq 0 \ \ \forall i : c_i = c_{min} \\ u^*_i = 0 \ \ \forall i : c_i \in (c_{min}, c_{min} + 2\gamma) \\ u^*_i = -\frac{1}{N} \ \ \forall i \in I_{big} \setminus S_2 \\ u^*_i = -\frac{1}{N} \ \ \forall i : c_i > c_{min} + 2\gamma \\ u^*_i = 0 \ \ \forall i \in S_1 \cup S_2 \end{array} \right\}$ . *Then*

$$conv\left(\cup_{S_1,S_2} U^*_{S_1,S_2}\right) = \arg\min_{u \in U} \sum_{i=1}^{N} (\frac{1}{N} + u_i) \cdot c_i + \gamma\|u\|_1. \tag{4}$$

127 The theorem explains that the construction of any optimal solution $u^*$ essentially reduces to cate-
128 gorizing each of the losses among $\{c_i = J(\theta; x_i, y_i)\}_{i=1}^{N}$ as "small" or "big", according to their
129 position in the partitioning of $[c_{min}, \infty) = [c_{min}, c_{min} + 2\gamma) \cup [c_{min} + 2\gamma, \infty)$. For losses that
130 occur at the break points of $c_{min}$ and $c_{min} + 2\gamma$, this classification can be arbitrary - hence, the use
131 of $S_1$ and $S_2$ set configurations to capture this degree of freedom.

132 In particular, those points with losses $c_i$ exceeding $c_{min} + 2\gamma$ are down-weighted to zero and ef-
133 fectively removed from the dataset. And in the event that $c_{max} - c_{min} < 2\gamma$, no loss reweighting
134 occurs. In this manner, while lasso produces sparse solutions in the model parameter space, RRM
135 produces sparse weight vectors by assigning zero weight to data points with high losses.

136 Consequently, if $\chi := \{i : c_i \in (c_{min} + 2\gamma, \infty)\}$ converges over the course of any algorithmic
137 scheme, e.g., Algorithm 1, to some set $C$, then we can conclude that these data points are effectively
138 removed from the dataset even if the training of $\theta$ might proceed. This convergence was observed in
139 the experiments of Section 6. It is hence of possible consideration to tune $\gamma$ for consistency with an
140 estimate $\alpha \in [0, 1]$ of labeling error in the dataset $\{(x_i, y_i)\}_{i=1}^{N}$. More precisely, we may tune $\gamma$ so
141 that $\frac{|\chi|}{N} \approx \alpha$.

## 142 3.4 RRM and Optimistic Wasserstein Distributionally Robust Optimization

143 In this section, we discuss RRM's relation to distributionally robust and optimistic optimization
144 formulations. Indeed, (3)'s formulation as a min-min problem bears resemblance to optimistic for-
145 mulations of recent works, e.g., [19]. We will see as well that the minimization in $u$, as considered
146 in Theorem 3.1, relates to an approximation of a data-driven Wasserstein Distributionally Robust
147 Optimization (DRO) formulation [30].

### 148 3.4.1 Loss-reweighting via Data-Driven Wasserstein Formulation

149 For this discussion, as it relates to reweighting, we will lift the feature-label space $\mathcal{Z} = \mathcal{X} \times \mathcal{Y}$.
150 More precisely, we let $\mathcal{W} := \mathbb{R}_+$ denote a space of *weights*. Next, we say $\mathcal{W} \times \mathcal{Z}$ has an unknown
151 probability distribution $\mathcal{D}$ such that $\pi_\mathcal{Z}\mathcal{D} = D$ and $\Pi_\mathcal{W}\mathcal{D}(\{1\}) = 1$. In words, all possible (w.r.t.
152 $D$) feature-label pairs have a weight of 1. Finally, we define an *auxiliary loss* $\ell : \mathcal{W} \times \mathcal{Z} \times \Theta$ by
153 $\ell(w, z; \theta) := w \cdot J(x, y; \theta)$, for any $z = (x, y) \in \mathcal{Z}$.

154 Given a sample $\{(1, x_i, y_i)\}_{i=1}^{N}$, just as in Section 3.2, we can opt not to take as granted the result-
155 ing empirical distribution $\mathcal{D}_N$ because of the possibility that $|C|$-many have incorrect labels (i.e.,
156 $y_i \neq \tilde{y}_i$). Instead, we will admit alternative distributions obtained by shifting the $\mathcal{D}_N$'s probability
157 mass off "corrupted" tuples $(1, x_i, y_i)_{i \in C}$ to possibly $(0, x_i, y_i)$, $(1, x_i, \tilde{y}_i)$, or even some other tuple
158 $(1, x_j, \tilde{y}_j)$ with $j \notin C$ for example - equivalently, eliminating, correcting, or replacing the sample,
159 respectively. In order to admit such favorable corrections to $\mathcal{D}_N$, we can consider the optimistic
160 [19, 30] data-driven problem

$$\min_\theta \left( v_N(\theta) := \min_{\tilde{\mathcal{D}}:W_1(\mathcal{D}_N,\tilde{\mathcal{D}})\leq\epsilon} \mathbb{E}_{\tilde{\mathcal{D}}}\left[\ell(w, z; \theta)\right] \right), \tag{5}$$

161 in which for each parameter tuning $\theta$, $v_N(\theta)$ measures the expected auxiliary loss with respect to
162 the most favorable distribution within an $\epsilon$ - prescribed $W_1$ (1- Wasserstein) distance of $\mathcal{D}_N$. It turns
163 out that a budgeted deviation of the weights alone (and not the feature-label pairs) can approximate
164 (up to an error diminishing in $N$) $v_N(\theta)$. More precisely, we derive the following approximation
165 along similar lines to [30].

**Proposition 3.2.** *Let $\epsilon > 0$, and suppose for any $\theta$, $\max_{(x,y)\in\mathcal{Z}} |J(\theta;x,y)| < \infty$. Then there exists $\kappa \geq 0$ such that for any $\theta$, the following problem*

$$v_N^{MIX}(\theta) := \min_{u_1,\ldots,u_N} \sum_{i=1}^N (\frac{1}{N} + u_i) \cdot J(\theta;x_i,y_i) + \gamma_\theta \sum_{i=1}^N |u_i|$$

$$s.t. \ u_i + \frac{1}{N} \geq 0 \ \ i = 1,\ldots,N$$

*satisfies $v_N(\theta) + \frac{\kappa}{N} \geq v_N^{MIX}(\theta) \geq v_N(\theta)$.*

*In particular, $-\gamma_\theta \leq \min_i J(\theta;x_i,y_i)$, and $\{i : J(\theta;x_i,y_i) > \gamma_\theta\}$ are all down-weighted to zero, i.e., $u_i^* = -\frac{1}{N}$ for any $u^*$ solving $v_N^{MIX}(\theta)$.*

In summary, while the optimistic Wasserstein formulation would permit correction to $\mathcal{D}_N$ with a combination of reweighting and/or feature-label revision, the above indicates that a process focused on reweighting alone could accomplish a reasonable approximation; further, upon comparison to (3), we see that RRM is a constrained version of this approximating problem, that is,

$$v(\theta) \geq v_N^{MIX}(\theta) \geq v_N(\theta).$$

Hence, in some sense, we can confirm that RRM is an optimistic methodology but that it is less optimistic than the data-driven Wasserstein approach.

## 3.5 RRM Algorithm

Towards solving problem (3) in the two decisions $\theta$ and $u$, we proceed iteratively with a block-coordinate descent heuristic outlined in Algorithm 1, whereby we update the two separately in cyclical fashion. In other words, we update $\theta$ while holding $u$ fixed, and we update $u$ whilst holding $\theta$ fixed. The update of $\theta$ is an SGD step on a batch of $s-$ many samples. The update of $u$ reduces to a linear program. In light of the discussion in 3.4, we also outline an Adversarial Rockafellian Relaxation method (A-RRM), an execution of RRM that includes a perturbation (parameterized by $\epsilon \geq 0$) to the feature $x$ of a sample $(x,y)$, for the purposes of adversarial training.

---

**Algorithm 1** (Adversarial) Rockafellian Relaxation Algorithm (A-RRM/RRM)

---

**Require:** Perturbation Multiplier $\epsilon \in [0,1]$, Number of epochs $\sigma$, Batch size $s \geq 1$, learning rate $\eta > 0$, regularization parameter $\gamma > 0$, reweighting step $\mu \in (0,1)$.
 $u \leftarrow 0 \in \mathbb{R}^N$
 **repeat**
   **for** $e = 1,\ldots,\sigma$ **do**
     **for** $b = 1,\ldots,\lceil \frac{N}{s} \rceil$ **do**
       $\{(x_i^b,y_i^b)\}_{i=1}^s \leftarrow$ Draw Batch of size $s$ from $\{(x_i,y_i)\}_{i=1}^N$
       **for** i = 1, ..., s **do**
         $x_i^b \leftarrow x_i^b + \epsilon \cdot sign\left(\nabla_x J(\theta;(x_i^b,y_i^b))\right)$
       **end for**
       $\theta \leftarrow \theta - \eta \sum_{i=1}^s \left(\frac{1}{N} + u_i\right) \cdot \nabla_\theta J(\theta;(x_i^b,y_i^b))$
     **end for**
   **end for**
   $u^* \leftarrow \min_{u\in U} \sum_{i=1}^N \left(\frac{1}{N} + u_i\right) \cdot J(\theta;x_i,y_i) + \gamma\|u\|_1$
   $u \leftarrow \mu u^* + (1-\mu)u$
 **until** Desired Validation Accuracy or Loss

---

The stepsize parameters $\mu, \eta$ and the regularization parameter $\gamma$ are hyper-parameters that may be tuned, or guided by the general discussions above in Section 3.3.

The RRM algorithm, in which $\epsilon = 0$, is meant for contexts in which only label corruption and no feature corruption occurs. The A-RRM algorithm, for which $\epsilon > 0$, is intended for contexts in which both label and feature corruption is anticipated.

## 4   Datasets

We select several datasets to evaluate RRM. In some cases, the selected dataset is nearly pristine. In these cases we perturb the dataset to achieve various types and levels of corruption. Other datasets consist of weakly labeled examples, which we maintain unaltered. The varied data domains and regimes of corruption enable a robust evaluation of RRM.

**MNIST** [17]: A multi-class classification dataset consisting of 70000 images of digits zero through nine. 60000 digits are set aside for training and 10000 for testing. 0%, 5%, 10%, 20%, and 30% of the training labels are swapped for different, randomly selected digits. The test set labels are unmodified.

**Toxic Comments** [6]: A multi-label classification problem from JIGSAW that consists of Wikipedia comments labeled by humans for toxic behavior. Comments can be any number (including zero) of six categories: toxic, severe toxic, obscene, threat, insult, and identity hate. We convert this into a binary classification problem by treating the label as either none of the six categories or at least one of the six categories. This dataset is a public dataset used as part of the Kaggle Toxic Comment Classification Challenge.

**IMDb** [18]: A binary classification dataset consisting of 50000 movie reviews each assigned a positive or negative sentiment label. 25000 reviews are selected randomly for training and the remaining are used for testing. 25%, 30%, 40%, and 45% of the labels of the training reviews are randomly selected and swapped from positive sentiment to negative sentiment, and vice versa, to achieve four training datasets of desired levels of label corruption. The test set labels are unmodified.

**Tissue Necrosis**: A binary classification dataset consisting of 7874 256x256-pixel hematoxylin and eosin (H&E) stained RGB images derived from [2]. The training dataset consists of 3156 images labeled non-necrotic, as well as 3156 images labeled necrotic. The training images labeled non-necrotic contain no necrosis. However, only 25% of the images labeled necrotic contain necrotic tissue. This type of label error can be expected in cases of weakly-labeled Whole Slide Imagery (WSI). Here, an expert pathologist will provide a slide-level label for a potentially massive slide consisting of gigapixels, but they lack time or resources to provide granular, segmentation-level annotations of the location of the pathology in question. Also, the diseased tissue often occupies a small portion of the WSI, with the remainder consisting of normal tissue. When the gigapixel-sized WSI is subsequently divided into sub-images of manageable size for typical machine-learning workflows, many of the sub-images will contain no disease, but will be assigned the "weak" label chosen by the expert for the WSI. The test dataset consists of 718 necrosis and 781 non-necrosis 256x256-pixel H&E images, which were also derived from [2]. For both the training and test images, [2] provide segmentation-level necrosis annotations, so we are able to ensure a pristine test set, and, in the case of the training set, we were able to identify the corrupted images for the purpose of algorithm evaluation.

## 5   Architectures

We do not strive to develop a novel NN architectures capable of defeating current state-of-the-art (SOA) performance in each data domain. Nor do we focus on developing *robust architectures* as described in [28]. Rather, we select a reasonable NN architecture and measure model performance with and without the application of RRM. This approach enables us to demonstrate the general superiority of RRM under varied data domains and NN architectures. We discuss the underlying NN architectures that we employ in this section.

**MNIST**: The MNIST dataset has been studied extensively and harnessed to investigate novel machine-learning methods, including CNNs [4]. We adopt a basic CNN architecture with a few convolutional layers. The first layer has a depth of 32, and the next two layers have a depth of 64. Each convolutional layer employs a kernel of size three and the ReLU activation function followed by a max-pooling layer employing a kernal of size 2. The last convolutional layer is connected to a classification head consisting of a 100-unit dense layer with ReLU activation, followed by a 10-unit dense layer with softmax activation. In total, there are 159254 trainable parameters. Categorical cross-entropy is employed for the loss function.

**Toxic Comments**: We use a simple model with only a single convolutional layer. A pretrained embedding from FastText is first used to map the comments into a 300 dimension embedding space, followed by a single convolutional layer with a kernel size of two with a ReLU activation layer followed by a max-pooling layer. We then apply a 36-unit dense layer, followed by a 6 unit dense layer with sigmoid activation. Binary cross-entropy is used for the loss function.

**IMDb**: Transformer architectures have achieved SOA performance on the IMDb dataset sentiment analysis task [7, 32]. As such, we a adopt a reasonable transformer architecture to assess RRM. We utilize the DistilBERT [25] architecture with low-rank adaptation (LoRA) [13] for large language models, which reduces the number of trainable weights from 67584004 to 628994. In this manner, we reduce the computational burden, while maintaining excellent sentiment analysis performance. Binary cross-entropy is employed for the loss function.

**Tissue Necrosis**: Consistent with the computational histopathology literature [21], we employ a convolutional neural network (CNN) architecture for this classification task. In particular, a ResNet-50 architecture with pre-trained ImageNet weights is harnessed. The classification head is removed and replaced with a dense layer of 512 units and ReLU activation function, followed by an output layer with a single unit using a sigmoid activation function. All weights, with the exception of the new classification head are frozen, resulting in 1050114 trainable parameters out of 24637826. Binary cross-entropy is employed for the loss function.

## 6 Experiments and Results

In this work, we have discussed errors/perturbations/corruption to features and labels. We now perform experiments to see how RRM performs under one or the other, or both. The MNIST experiments are performed under a setting of both adversarial perturbation, as well as label corruption. The Toxic Comments experiments are performed under settings of label corruption only. All experiments are performed using a combination of GPU resources, both cloud-base, as well as access to an on-premise high-performance computing (HPC) facility. We refer the reader to the Appendix (Sections 6.3 and 6.4) for the experiments on IMDb and Tissue Necrosis.

### 6.1 MNIST

Twenty percent of the training data is set aside for validation purposes. Using Tensorflow 2.10 [1], 50 iterations of RRM are executed with $\sigma = 10$ epochs per iteration for a total of 500 epochs for a given hyperparameter setting. For RRM, the hyperparameter settings of $\mu$ and $\gamma$ at 0.5 and 2.0, respectively, are based on a search to optimize validation set accuracy. For contrast, we perform a comparable 500 epochs using ERM. Both ERM and RRM employ stochastic gradient descent (SGD) with a learning rate ($\eta$) of 0.1. Each time a batch is drawn, each training image is perturbed using the Fast Gradient Sign Method (FGSM) [11] adversarial attack: $adv_x = x + \epsilon \cdot sign(\nabla_x J(\theta, x, y))$, where $adv_x$ is the resulting perturbed image, $x$ is the original image, $y$ is the image label, $\epsilon$ is a multiplier controlling the magnitude of the image perturbation, $\theta$ are the model parameters, and $J$ is the loss. An $\epsilon = 1.0$ is used for all training image perturbations.

For each of the 0%, 5%, 10%, 20%, and 30% training label corruption levels, we compare adversarial training (AT) and adversarial RRM (A-RRM) performance under varios regimes of test set perturbation ($\epsilon_{test} \in 0.0, 0.1, 0.25, 0.5, 1.0$). In Table 1 we show the test set accuracy achieved when validation set accuracy peaks. We can see that training with an $\epsilon_{train} = 1.0$ and testing with lower $\epsilon_{test}$ levels of $0.00, 0.10$, and $0.25$, results in a drastic degradation in accuracy for AT for corruption levels greater than 0%. This performance collapse is not observed when using A-RRM. Given that it may be difficult to anticipate the adversarial regime in production environments, A-RRM seems to confer a greater benefit than AT.

We examine the $u_i$-value associated with each training observation, $i$, from iteration-to-iteration of the heuristic algorithm. Table 2 summarizes the progression of the $u_i$-vector across its 49 updates for the dataset corruption level of 20%. Column "1. iteration" shows the distribution of $u_i$-values following the first u-optimization for both the 9600 corrupted training observations and the 38400 clean training observations. Initially, all $u_i$-values are approximately equal to 0.0. It is once again observed that, over the course of iterations, the $u_i$-values noticeably change. In column "10. iteration" it can be seen that a significant number of the $u_i$-values of the corrupted training observations

Table 1: Test accuracy (%) for AT and A-RRM on MNIST under different levels of corruption $C$ and test-set adversarial perturbation $\epsilon_{test}$.

| | Percentage Corrupted Training Data | | | | | | | | | |
|---|---|---|---|---|---|---|---|---|---|---|
| $C$ | 0% | | 5% | | 10% | | 20% | | 30% | |
| $\epsilon_{test}$ | AT | A-RRM | AT | A-RRM | AT | A-RRM | AT | A-RRM | AT | A-RRM |
| 0.00 | **97** | 96 | 63 | **95** | 57 | **97** | 58 | **96** | 26 | **86** |
| 0.10 | **95** | 93 | 64 | **92** | 71 | **94** | 61 | **93** | 20 | **82** |
| 0.25 | **93** | 90 | 83 | **91** | 88 | **92** | 84 | **90** | 74 | **81** |
| 50 | **91** | 88 | **94** | 91 | **94** | 90 | **90** | 88 | **97** | 80 |
| 1.00 | **86** | 83 | **95** | 90 | **94** | 86 | **88** | 83 | **98** | 77 |

achieve negative values, while a large majority of the $u_i$-values for the clean training observations remain close to 0.0. Finally, column "49. iteration" displays the final $u_i$-values. 9286 out of 9600 of the corrupted training observations have achieved a $u_i \in (-2.08, -1.56] \cdot 10-5$. This means these training observations are removed, or nearly-so, from consideration because this value cancels the nominal probability $1/N = 2.08 \cdot 10-5$. It is observed that a large majority (35246/38400) clean training observations remain with their nominal probability. This helps explain the performance benefit of A-RRM over AT. A-RRM "removes" the corrupted data points in-situ, whereas AT does not. It appears that under adversarial training regimes with corrupted training data, it is essential to identify and "remove" the corrupted examples, especially if the level adversarial perturbation encountered in the test set is unknown, or possibly lower than the level of adversarial perturbation applied to the training set.

Table 2: Evolution of u-vector across 9600 corrupted data points and 38400 clean data points. Note that $1/(9600 + 38400) = 2.08 \cdot 10-5$.

| | 1. iteration | | 10. iteration | | 49. iteration | |
|---|---|---|---|---|---|---|
| $u_i$ value | corrupted data points | clean data points | corrupted data points | clean data points | corrupted data points | clean data points |
| $\gg 0$ | 0 | 1 | 0 | 4. | 0 | 25 |
| $\approx 0$ | 8844 | 38385 | 2058 | 37524 | 91 | 35246 |
| $(-0.52, 0.00) \cdot 10-5$ | 0 | 0 | 7 | 36 | 146 | 1655 |
| $(-1.04, -0.52] \cdot 10-5$ | 0 | 0 | 41 | 45 | 43 | 155 |
| $(-1.56, -1.04] \cdot 10-5$ | 756 | 14 | 415 | 174 | 34 | 168 |
| $(-2.08, -1.56] \cdot 10-5$ | 0 | 0 | 7079 | 617 | 9286 | 1151 |

## 6.2 Toxic Comment

We use the Toxic Comment dataset to test the efficacy of RRM on low prevalence text data. The positive (toxic) comments consist of only 3% of the data and we corrupt anywhere from 1% to 20% of the labels. There are a total of 148,000 samples, and we set aside 80% for training and 20% for test. $\sigma = 2$ with 3 iterations of the heuristic algorithm results in a total of 6 epochs, and ERM is run for a total of 6 epochs to make the results comparable. Since the data is highly imbalanced, we look at the area under the curve of the precision/recall curve to assess the performance of the models. Unsurprisingly, as the noise increase, the model performance decreases. We note that RRM outperforms ERM across all noise levels tested, though as the noise increase, the gap between RRM and ERM decreases.

Table 3: Comparison of training and test area under the precision/recall curve for ERM and RRM at noise levels ranging from 1% to 20%.

| Method | Percentage Corrupted Training Data | | | | | |
|---|---|---|---|---|---|---|
| | 1% | 5% | 7% | 10% | 15% | 20% |
| ERM (train) | 0.2904 | 0.2006 | 0.1589 | 0.1302 | 0.1073 | 0.0920 |
| RRM (train) | 0.6875 | 0.4458 | 0.3805 | 0.3087 | 0.2438 | 0.1966 |
| ERM (test) | 0.5861 | 0.3970 | 0.3246 | 0.2550 | 0.2013 | 0.1717 |
| RRM (test) | **0.6705** | **0.4338** | **0.3619** | **0.2824** | **0.2208** | **0.1861** |

 **6.3 IMDb**

Twenty percent of the training data is set aside for validation purposes. Using Pytorch 2.1.0 [3], 30 iterations of RRM are executed, with $\sigma = 10$ epochs per iteration for a total of 300 epochs for a given hyperparameter setting. For RRM, the hyperparameter settings of $\mu$ and $\gamma$ at 0.5 and 0.4, respectively, are based on a search to optimize validation set accuracy. For contrast, we perform a comparable 300 epochs using ERM. Both ERM and RRM employ stochastic gradient descent (SGD) with a learning rate ($\eta$) of 0.001. In Table 4 we record both the test set accuracy achieved when validation set accuracy peaks, as well as the maximum test set accuracy. At these high levels of corruption RRM consistently achieves a better maximum test set accuracy.

Table 4: Test accuracy (%) for ERM and RRM on IMDb under different levels of corruption. Test set accuracy at peak validation accuracy and maximum test set accuracy are recorded.

| Method | Percentage Corrupted Training Data | | | |
|--------|------|------|------|------|
|        | 25%  | 30%  | 40%  | 45%  |
| ERM    | *90.2*, 90.2 | 89.5, 89.6 | 86.4, 86.6 | *80.7*, 81.1 |
| RRM    | 90.1, **90.4** | *90.2*, **90.4** | *88.4*, **88.7** | 76.9, **82.6** |

**6.4 Tissue Necrosis**

Twenty percent of the training data is set aside for validation purposes, including hyperparameter selection. 60 iterations of RRM are executed, with $\sigma = 10$ epochs per iteration, for a total of 600 epochs for a given hyperparameter setting. For RRM, the hyperparameter settings of $\mu$ and $\gamma$ at 0.5 and 0.016, respectively, are based on a search to optimize validation set accuracy. For contrast, we perform a comparable 600 epochs using ERM. Both ERM and RRM employ stochastic gradient descent (SGD) with a learning rate ($\eta$) of 5.0 and 1.0, respectively. RRM achieves a test set accuracy at peak validation accuracy of **74.6**, and a maximum test set accuracy **77.2**, whereas ERM achieves 71.7 and 73.2, respectively. RRM appears to confer a performance benefit under this regime of weakly labeled data.

## 7 Conclusion

In this study, we demonstrate the robustness of the A-RRM algorithm in a variety of data domains, data corruption schemes, model architectures and machine learning applications. In the MNIST example we show that conducting training in preparation for deployment environments with varied levels of adversarial attacks, one can benefit from implementation of the A-RRM algorithm. This can lead to a model more robust across levels of both feature perturbation and high levels of label corruption. We also demonstrate the mechanism by which A-RRM operates and confers superior results: by automatically identifying and removing the corrupted training observations at training time execution.

The Toxic Comment example presents another challenging classification problem, characterized by a low prevalence target class amidst label noise. Our experiments demonstrate that as the amount of label noise increases, standard methods become increasingly ineffective. However, RRM remains reasonably robust under varying degrees of label corruption. Therefore, RRM could be a valuable addition to the set of tools being developed to enhance the robustness of AI-based decision engines.

In the IMDb example we demonstrate that RRM can confer benefits to the sentiment analysis classification task using pre-trained large models under conditions of high label corruption. The success of fine-tuning in LLMs depends, in large part, on access to high quality training examples. We have shown that RRM can mitigate this need by allowing effective training in scenarios of high training data corruption. As such, resource allocation dedicated to dataset curation may be lessened by the usage of RRM.

In the Tissue Necrosis example, we demonstrate that RRM also confers accuracy benefits to the necrosis identification task provided weakly labeled WSIs. Again, RRM can mitigate the need for expert-curated, detailed pathology annotations, which are costly and time-consuming to generate.

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

# A   Appendix / supplemental material

## A.1   Section 3 Proofs

**Theorem 3.1.** *Let $\gamma > 0$ and $c = (c_1, \ldots, c_N) \in \mathbb{R}^N$, with $c_{min} := \min_i c_i$, and $c_{max} := \max_i c_i$. Write $I_{min} := \{i : c_i = c_{min}\}$, $I_{big} := \{i : c_i = c_{min} + 2\gamma\}$, and for any $S_1 \subseteq I_{min}, S_2 \subseteq I_{big}$, define the polytope*

$$U^*_{S_1,S_2} := \left\{ u^* \in U : \begin{array}{c} u_i^* \geq 0 \ \forall i : c_i = c_{min} \\ u_i^* = 0 \ \forall i : c_i \in (c_{min}, c_{min} + 2\gamma) \\ u_i^* = -\frac{1}{N} \ \forall i \in I_{big} \setminus S_2 \\ u_i^* = -\frac{1}{N} \ \forall i : c_i > c_{min} + 2\gamma \\ u_i^* = 0 \ \forall i \in S_1 \cup S_2 \end{array} \right\}. \text{ Then}$$

$$conv\left( \cup_{S_1,S_2} U^*_{S_1,S_2} \right) = \arg\min_{u \in U} \sum_{i=1}^{N} (\frac{1}{N} + u_i) \cdot c_i + \gamma \|u\|_1. \tag{4}$$

*Proof.* For any set $C$, let $\iota_C(x) = 0$ and $\iota_C(x) = \infty$ otherwise. We recognize that $u^\star$ is a solution of the minimization problem if and only if it is a minimizer of the function $h$ given by

$$h(u) = \sum_{i=1}^{N} \left( c_i/N + u_i c_i + \gamma |u_i| + \iota_{[0,\infty)}(1/N + u_i) \right) + \iota_{\{0\}}\left( \sum_{i=1}^{N} u_i \right)$$

Thus, because $h(u) > -\infty$ for all $u \in \mathbb{R}^N$ and $h$ is convex, $u^\star$ is a solution of the minimization problem if and only if $0 \in \partial h(u^\star)$ by Theorem 2.19 in [24]. We proceed by characterizing $\partial h$.

Consider the univariate function $h_i$ given by

$$h_i(u_i) = c_i/N + u_i c_i + \gamma |u_i| + \iota_{[0,\infty)}(1/N + u_i).$$

For $u_i \geq -1/N$, the Moreau-Rockafellar sum rule (see, e.g, [24, Theorem 2.26]) gives that

$$\partial h_i(u_i) = c_i + \begin{cases} \{\gamma\} & \text{if } u_i > 0 \\ [-\gamma, \gamma] & \text{if } u_i = 0 \\ \{-\gamma\} & \text{if } -1/N < u_i < 0 \\ (-\infty, -\gamma] & \text{if } u_i = -1/N. \end{cases}$$

For $u = (u_1, \ldots, u_N) \in [-1/N, \infty)^N$, we obtain by Proposition 4.63 in [24] that

$$\partial\Big(\sum_{i=1}^N h_i\Big)(u) = \partial h_1(u_1) \times \cdots \times \partial h_N(u_N).$$

Let $h_0$ be the function given by $h_0(u) = \iota_{\{0\}}(\sum_{i=1}^N u_i)$. Again invoking the Moreau-Rockafellar sum rule while recognizing that the interior of the domain of $\sum_{i=1}^N h_i$ intersects with the domain of $h_0$, we obtain

$$\partial h(u) = \partial\Big(\sum_{i=1}^N h_i\Big)(u) + \partial h_0(u) = \partial h_1(u_1) \times \cdots \times \partial h_N(u_N) + \begin{bmatrix} 1 \\ \vdots \\ 1 \end{bmatrix} \mathbb{R}$$

for any $u = (u_1, \ldots, u_N)$ with $u_i \geq -1/N$, $i = 1, \ldots, N$, and $\sum_{i=1}^N u_i = 0$. Hence, $u^* \in U$ is optimal if and only if for some $\lambda \in \mathbb{R}$,

$$\lambda \in \begin{cases} \{c_i + \gamma\} & \text{if } u_i^\star > 0 \\ [c_i - \gamma, c_i + \gamma] & \text{if } u_i^\star = 0 \\ \{c_i - \gamma\} & \text{if } u_i^\star \in (-1/N, 0) \\ (-\infty, c_i - \gamma] & \text{if } u_i^\star = -1/N. \end{cases}$$

It follows that $\lambda = c_{min} + \gamma$ can accompany any optimal $u^*$ in satisfying the above; hence, the result follows.

$\square$

**Proposition A.1.** *Let $\epsilon > 0$, and suppose for any $\theta$, $\max_{(x,y)\in\mathcal{Z}} |J(\theta; x, y)| < \infty$. Then there exists $\kappa \geq 0$ such that for any $\theta$, the following problem*

$$v_N^{MIX}(\theta) := \min_{u_1, \ldots, u_N} \sum_{i=1}^N \Big(\frac{1}{N} + u_i\Big) \cdot J(\theta; x_i, y_i) + \gamma_\theta \sum_{i=1}^N |u_i|$$

$$s.t. \ u_i + \frac{1}{N} \geq 0 \ \ i = 1, \ldots, N$$

*satisfies $v_N(\theta) + \frac{\kappa}{N} \geq v_N^{MIX}(\theta) \geq v_N(\theta)$.*

*In particular, $-\gamma_\theta \leq \min_i J(\theta; x_i, y_i)$, and $\{i : J(\theta; x_i, y_i) > \gamma_\theta\}$ are all down-weighted to zero, i.e., $u_i^* = -\frac{1}{N}$ for any $u^*$ solving $v_N^{MIX}(\theta)$.*

*Proof.* Fix $\theta$. Then for any $z = (x, y) \in \mathcal{Z}$, the function $\ell(\cdot, z, \theta)$ is linear, and hence Lipschitz with constant $\ell(1, z, \theta) = J(\theta; x, y) \leq \max_{(x,y)\in\mathcal{Z}} |J(\theta; x, y)| < \infty$.

By Lemma 3.1 of [30] and/or Corollary 2 of [9],

$$v_N^{MIX}(\theta) := \min_{\tilde{w}^1, \ldots, \tilde{w}^N \geq 0} \frac{1}{N} \sum_{i=1}^N \ell(\tilde{w}^i, z^i; \theta)$$

$$s.t. \ \frac{1}{N} \sum_{i=1}^N |\tilde{w}^i - w^i| \leq \epsilon$$

provides the stated approximation of $v(\theta)$.

504 Upon introducing the change of variable $u_i = \frac{\tilde{w}^i}{N} - \frac{1}{N}$, and applying a Lagrange multiplier $\gamma_\theta$ to
505 the $\epsilon-$ budget constraint (any convex dual optimal multiplier), we recover

$$\min_{u_1,\ldots,u_N} \sum_{i=1}^{N} \ell(u_i + \frac{1}{N}, z^i; \theta) + \gamma_\theta \sum_{i=1}^{N} |u_i|$$

$$\text{s.t. } u_i + \frac{1}{N} \geq 0 \ \ i = 1, \ldots, N$$

506 $\square$

