# OpenReview forum: "Mitigating the Impact of Labeling Errors on Training via Rockafellian Relaxation"
_NeurIPS.cc/2024/Conference — Submitted to NeurIPS 2024_

### Official Review · Reviewer_yNxu · 2024-07-08

**Soundness:** 3
**Presentation:** 3
**Contribution:** 3
**Rating:** 5
**Confidence:** 3

**Summary:**

This paper proposes a methodology called Rockafellian Relaxation (RR) to mitigate the impact of labeling errors in neural network training. The method is architecture-independent and integrates concepts from adversarial training to address dataset imperfections robustly. Through theoretical justifications and a series of experiments on standard datasets like MNIST and Toxic Comments, the paper demonstrates that RR can significantly improve the performance of neural networks trained under various corruption levels. The paper’s contributions are particularly valuable as they provide a new tool for improving training accuracy in the presence of label noise, enhancing the robustness and applicability of machine learning models in diverse and error-prone real-world settings.

**Strengths:**

Originality: The paper's approach to using Rockafellian Relaxation for addressing labeling errors is innovative, especially the combination with adversarial training concepts.

Quality: The method is grounded in solid theoretical justification, and the empirical results show marked improvements over existing methods.

Clarity: The explanations of the methodologies and the algorithms are clear and detailed, making it easier to understand the operational aspects of the proposed solution.

Significance: The significance of this work lies in its potential to improve training robustness across various domains and dataset imperfections, which is highly relevant for deploying machine learning models in error-prone real-world environments.

**Weaknesses:**

Computational Complexity: The added complexity might limit the practical application of the method in scenarios with constrained computational resources.

**Questions:**

Scalability: How does the RRM scale in terms of computational cost and effectiveness with larger and more complex datasets?

**Limitations:**

Generalization to Different Noise Types: While the method is tested against uniform label noise, its effectiveness against other types of noise is not thoroughly investigated.

Dependence on Hyperparameter Tuning: The effectiveness of RRM is likely sensitive to the choice of hyperparameters, such as the regularization term and the parameters controlling the adversarial component. The paper does not provide extensive guidance on hyperparameter selection, which could affect the reproducibility and ease of application in different scenarios.

---

> ### Author Rebuttal · Authors · 2024-08-06
>
> ## **Question**
> Scalability: How does the RRM scale in terms of computational cost and effectiveness with larger and more complex datasets?\\
> ## **Rebuttal**
> Each iteration of RRM, as outlined in Algorithm 1 on page 5, is comprised of two tasks: (1) a gradient step and (2) a poly-size (in training data size) linear program that constitutes loss-reweighting. As task (1)'s gradient step is entirely standard practice, and task (2)'s complexity is poly-time in the training data size, the scaling of computation with larger datasets is not limiting in nature. As a side note, the linear program of (2) has a structure that can be exploited for fast computation. We are happy to add such a discussion on these complexity matters, if given the opportunity.
>
>
> ## **Question**
> Generalization to Different Noise Types: While the method is tested against uniform label noise, its effectiveness against other types of noise is not thoroughly investigated.
> ## **Rebuttal**
> Although we explicitly state the use of uniform label noise in Section 3.1, which is indeed a very common scheme in the literature, we clarify that our analysis in fact did not rely on this assumption. Towards providing you insight into the non-uniform case, we have repeated the experiments on MNIST that produced Table 1, but now with non-uniform label noise. More precisely, after uniformly randomly selecting $C$ percent of the training pairs, we proceed to contaminate the label $y_i$ in each pair $(x_i, y_i)$ in the following non-uniform manner, as outlined below in the transition kernel matrix of (True Label, Contaminated Label) entries. For example, as the matrix below indicates, if the true label $y_i = 5$, then instead of uniformly randomly drawing an alternative digit $\tilde{y}_i$ from among $\{0, 1, \ldots, 9\} \setminus \{5\}$, we have
> $\tilde{y}_i =$ \
> 0 w.p. 0.051\
> 1 w.p. 0.017 \
> 2 w.p.  0. \
> 3 w.p.  0.627 \
> $\ldots$
> | True \ Contaminated | 0     | 1     | 2     | 3     | 4     | 5     | 6     | 7     | 8     | 9     |
> |---------------------|-------|-------|-------|-------|-------|-------|-------|-------|-------|-------|
> | 0                   | 0     | 0.077 | 0.077 | 0.154 | 0     | 0.077 | 0.385 | 0     | 0.154 | 0.077 |
> | 1                   | 0     | 0     | 0.333 | 0.111 | 0     | 0.111 | 0.111 | 0     | 0.333 | 0     |
> | 2                   | 0.097 | 0.065 | 0     | 0.258 | 0.032 | 0     | 0.097 | 0.194 | 0.258 | 0     |
> | 3                   | 0     | 0     | 0.125 | 0     | 0     | 0.125 | 0     | 0.125 | 0.625 | 0     |
> | 4                   | 0.111 | 0.037 | 0.074 | 0.074 | 0     | 0.074 | 0.222 | 0.037 | 0.111 | 0.259 |
> | 5                   | 0.051 | 0.017 | 0     | 0.627 | 0.017 | 0     | 0.153 | 0     | 0.102 | 0.034 |
> | 6                   | 0.235 | 0.176 | 0.059 | 0.059 | 0.059 | 0.176 | 0     | 0     | 0.235 | 0     |
> | 7                   | 0.050 | 0.225 | 0.200 | 0.125 | 0     | 0     | 0     | 0     | 0.200 | 0.200 |
> | 8                   | 0.107 | 0.036 | 0.107 | 0.357 | 0.107 | 0.071 | 0.071 | 0.107 | 0     | 0.036 |
> | 9                   | 0.064 | 0.170 | 0     | 0.213 | 0.170 | 0.170 | 0.021 | 0.085 | 0.106 | 0     |
>
> These entries were generated by the confusion matrix of an imperfect MNIST classifier.
>
> The results from this new experiment confirm the performance benefits that were observed (compare to Table 1) under conditions of uniform label contamination.
>
> | $\epsilon_{test}$\C | 0% AT | 0% A-RRM | 5% AT | 5% A-RRM | 10% AT | 10% A-RRM | 20% AT | 20% A-RRM | 30% AT | 30% A-RRM |
> |-|-|-|-|-|-|-|-|-|-|-|
> | 0 | 96.5 | **97.3** | 93.6 | **95.6** | 60.4 | **87.8** | 32.2 | **92.4** | 58.3 | **89.2** |
> | 0.1  | 93.4 | **95.2** | 89.3 | **92.4** | 63.2 | **84.7** | 42.5 | **89.5** | 56.1 | **81.7** |
> | 0.25 | 92.4 | **93.1** | 87.9 | **90.6** | **86.9** | 86.3 | 80.6 | **89.3** | 69.0 | **79.8** |
> | 0.5 | **92.0** | 90.9 | **90.3** | 89.8 | **94.4** | 91.2 | **92.9** | 89.2 | **85.2** | 81.6 |
> | 1 | **89.4** | 85.5 | **90.3** | 86.8 | **94.9** | 92.6 | **93.9** | 86.6 | **81.8** | 77.8 |
>
> ## **Question**
> Dependence on Hyperparameter Tuning: The effectiveness of RRM is likely sensitive to the choice of hyperparameters, such as the regularization term and the parameters controlling the adversarial component. The paper does not provide extensive guidance on hyperparameter selection, which could affect the reproducibility and ease of application in different scenarios.
>
> ## **Rebuttal**
> Qualitatively, across the four diverse scenarios that we experimented, we found the hyper-parameters search non-burdensome, and yielded beneficial RRM performance. In particular, we follow standard practice of leveraging a validation set for hyperparameter selection. In the discussion following Theorem 3.1, we provide insight into how $\gamma$ may be tuned in relation to an estimate $\alpha$ of the labeling error in the dataset. If invited we can quantify more precisely the sensitivity in performance changes due to hyper-parameter changes.

---

### Official Review · Reviewer_TFmX · 2024-07-12

**Soundness:** 3
**Presentation:** 2
**Contribution:** 3
**Rating:** 5
**Confidence:** 3

**Summary:**

This work proposes a loss reweighting scheme to train models in the presence of label errors. When training an NN with empirical risk minimization in this setting, one would want to assign a weight of zero to all datapoints that are mislabeled and a weight of one to all datapoints that are correctly labeled. This paper presents an automated method for accomplishing this weighting, called the Rockafellian Relaxation Method (RRM). It is noted in Theorem 3.1 that the inner minimization objective of RRM reduces to a linear programming problem, despite RRM being non convex in general. After relating RRM to distributionally robust optimization techniques, the adversarial variant of RRM is introduced (A-RRM), which includes adversarial perturbations to induce adversarial training as well as loss reweighting. Experiments on four datasets show that RRM and A-RRM outperform other methods in both adversarial settings and settings with high proportions of noisy labels.

**Strengths:**

This work addresses two different types of robustness: robustness to label noise and robustness to adversarial feature perturbation. It should be of interest to those who are generally interested in robust and trustworthy machine learning. Furthermore, the proposed training method has strong theoretical foundations, and its relation to other optimization formulations is discussed in detail. The theoretical results are validated in experiments that cover different datasets and types of data corruption.

**Weaknesses:**

The experimental section lacks a relevant baseline for comparison. As it stands, it is unclear how this compares to other noise-reduction techniques. The relationship to other techniques is discussed in the related work section, it would be nice if the purported benefits of this approach were borne out empirically.

The introduction of adversarial training in section 3.5 is under-motivated. Based on the earlier sections, it is unclear how label and adversarial feature corruptions are related to each other, why we would want to achieve robustness to both, and whether previous approaches have attempted this before. I would suggest explicitly motivating this earlier in the paper.

**Questions:**

- Could you expand on what you mean by RRM producing sparse weight vectors? Does assigning zero weight to data points with high losses result in sparsity in the parameter space?
- What is the reason for choosing the FGSM attack in A-RRM rather than a different attack, like PGD? Could other attacks be used in the place of FGSM?
- In Table 1, is the epsilon in the fourth row supposed to be 0.50 instead of 50?
- Do you have a hypothesis for why the test accuracy increases for models trained with AT and tested with $\epsilon_{test} = 0.50$ or $1.0$ when the level of corruption is increased (i.e. the AT columns in the bottom two rows of Table 1)? That seems like an unexpected result that may warrant further study?

**Limitations:**

The limitations are briefly discussed in the paper. As noted above, one main limitation is that it only studies $\ell_\infty$ bounded FGSM attacks. Furthermore, this paper only considers the uniform label noise model, and does not consider the case when label corruption might be correlated with features.

---

> ### Author Rebuttal · Authors · 2024-08-06
>
> ## **Question**
> Could you expand on what you mean by RRM producing sparse weight vectors? Does assigning zero weight to data points with high losses result in sparsity in the parameter space?
>
> ## **Response**
> In equation (3), the expressions $(\frac{1}{N} + u_i)$ are to be understood as the weight given to the $i$-th training sample. Hence, if we consider the weight vector $(\frac{1}{N} + u_i)_{i=1}^N$, it is "sparse" if there are many $u_i$ set to $-\frac{1}{N},$ equiv., when the weight vector carries many zero-value entries. Thus, by our comment that "RRM produces sparse weight vectors by assigning zero weight to data points with high losses", we mean to say that those samples that present sufficiently high losses are removed from consideration.
>
> We may have caused confusion surrounding "sparsity in the parameter space" with our statement: "...while lasso produces sparse solutions in the model parameter space, RRM produces sparse weight vectors by assigning zero weight to data points with high losses." In truth, sparsity in the (model) parameter(s) $\theta$ was not something we observed. Our intent was to contrast the sparsity in the weight vectors  $(\frac{1}{N} + u_i)_{i=1}^N$ of RRM with the sparsity that occurs in the model parameters of lasso-regularized, linear regression (i.e. the linear coefficients).
>
>
> ## **Question**
> What is the reason for choosing the FGSM attack in A-RRM rather than a different attack, like PGD? Could other attacks be used in the place of FGSM?
>
> ## **Response**
> Were it not for the 9-page limit, we would have catalogued more attacks. Certainly, attacks like PGD could be used in place of FGSM. A good direction to take for follow-on work!
>
> ## **Question**
> In Table 1, is the epsilon in the fourth row supposed to be 0.50 instead of 50?
>
> ## **Response**
> Yes, thanks for the catch!
>
> ## **Question**
> Do you have a hypothesis for why the test accuracy increases for models trained with AT and tested with $\epsilon_{test} = 0.50$ or $1.0$ when the level of corruption is increased (i.e. the AT columns in the bottom two rows of Table 1)? That seems like an unexpected result that may warrant further study?
>
> ## **Response**
> Upon scanning the $\epsilon_{test} = 1.00$ row of Table 1, we see that the AT test accuracy numbers read from left-to-right (corresponding to increasing corruption level) as 86, 95, 94, 88, 98. We're not certain that this necessarily indicates that the test accuracy is increasing with corruption level. We have a similar perspective in the case of $\epsilon_{test} = 0.50$.
>
> However, for some possible explanations of AT's perhaps unexpected performance$\ldots$
> - Stochastic nature of model training (e.g. shuffling the data, initializing of parameters, number of iterations, etc.)
> - The training perturbation level $\epsilon_{train} = 1$, so perhaps the closer $\epsilon_{test}$ gets to 1, the more similar the training and testing environments become, perhaps explaining the increase in AT accuracy with increasing $\epsilon_{test}$. We certainly see that conversely, as $\epsilon_{test}$ decreases to 0, AT's accuracy relative to RRM plummets.  Given that it may be difficult to anticipate the test perturbation $\epsilon_{test}$, RRM can provides a perturbation-robust alternative to AT.

---

> > ### Comment · Reviewer_TFmX · 2024-08-11
> >
> > Thank you for your response to my questions. I appreciate the empirical comparison to ELR in the response to reviewer UZpK. However, I still believe that a more thorough comparison is needed, as these results are fairly inconclusive and you state that a more thorough hyperparameter search is necessary. I will therefore be keeping my current score.

---

> > > ### Author Response · Authors · 2024-08-12
> > > **On the Matter of ELR versus RRM-wrapped ELR**
> > >
> > > Towards addressing your concern over the inconclusiveness of our comparison, due to a lack of hyperparameter search, we have endeavored to revisit the experiments, now with a hyperparameter search, to show how RRM-wrapping a loss methodology can enhance performance. The table below reports the test accuracy results obtained upon revisiting the experiments on MNIST3 with a hyperparameter search for all methods.
> > >
> > > |   |  |  |  |
> > > |----------|----------------------|----------------------|----------------------|
> > > |  **Method** \ **Contamination Level**         | 0.55                 | 0.60                 | 0.65                 |
> > > | ERM      |  0.90  |  0.77   |  0.46  |
> > > | RRM(ERM) |  **0.98**  |  **0.96**   |  **0.67**  |
> > > | ELR      |  0.98  |  0.97   |  0.82  |
> > > | RRM(ELR) |  **0.99**  |  **0.98**   |  **0.87**  |
> > >
> > > We highlight that the RRM-wrapped ERM method was uniformly better than ERM across all contamination levels examined. We observed this uniform enhancement for the case of ELR as well. Of particular note, once we tuned **ELR**'s hyperparameters, those same hyperparameter settings were maintained in **RRM(ELR)** (along with the RRM-specific hyperparameters tuned).
> > >
> > > **These results suggest that even after hyperparameter tuning of a method like ERM or ELR, there are further enhancements to be obtained with RRM-wrapping.**

---

### Official Review · Reviewer_UZpK · 2024-07-16

**Soundness:** 3
**Presentation:** 2
**Contribution:** 3
**Rating:** 5
**Confidence:** 4

**Summary:**

The paper presents Rockafellian Relaxation (RR), a new method to address labeling errors in machine learning datasets. RR is a loss reweighting technique that enhances neural network robustness against labeling errors and adversarial attacks, working across various data domains and model architectures. The key contribution is an approach that mitigates label corruption and class imbalance without needing clean validation sets, offering a practical solution for training robust models.

**Strengths:**

- The paper introduces Rockafellian Relaxation (RR), a novel loss reweighting methodology that addresses learning with noisy label problems

- The authors provide a solid theoretical basis for RR, relating it to optimistic and robust distributional optimization formulations. RR is also designed to be architecture-independent, making it a versatile tool applicable across different neural network architectures.

- The method does not rely on having clean validation data, which is of advantage in many real-world applications.

**Weaknesses:**

- While not explicitly mentioned, the iterative nature of the RR algorithm could potentially be computationally intensive, especially for large datasets.

- The method assumes a specific model of label noise (e.g., uniform label noise), which may not hold in all real-world scenarios.

- The paper could benefit from a more comprehensive comparison with other state-of-the-art methods for handling noisy labels, such as GCE [1], ELR[2], to better position RR in the existing literature.

[R1] Generalized Cross Entropy Loss for Training Deep Neural Networks with Noisy Labels

[R2] Early-Learning Regularization Prevents Memorization of Noisy Labels

**Questions:**

- Could the authors provide insights into the computational complexity of the RR algorithm, particularly in the context of large-scale datasets such as clothing1m?

- How does the performance of RR compare under different models of label noise, especially those that deviate from the assumed uniform label noise model such as asymmetric/instance-dependent label noise?

- How does RR perform relative to other state-of-the-art methods like Generalized Cross Entropy (GCE) and Early-Learning Regularization (ELR) in terms of handling noisy labels?

- How sensitive is the performance of RR to the choice of hyperparameters, and are there any techniques to optimize these selections effectively?

**Limitations:**

Authors have adequately addressed the limitations.

---

> ### Author Rebuttal · Authors · 2024-08-06
>
> # Question
> Could the authors provide insights into the computational complexity of the RR algorithm...?
> # Response
> Each iteration of RRM, as outlined in Algorithm 1 on page 5, is comprised of two tasks: (1) a gradient step and (2) a poly-size (in training data size) linear program that constitutes loss-reweighting. As task (1)'s gradient step is entirely standard practice, and task (2)'s complexity is poly-time in the training data size, the scaling of computation with larger datasets is not limiting in nature. As a side note, the linear program of (2) has a structure that can be exploited for fast computation. We are happy to add such a discussion on these complexity matters, if given the opportunity.
> # Question
> How does the performance of RR compare under different models of label noise, ...such as asymmetric/instance-dependent label noise?
> # Response
> Although we explicitly state the use of uniform label noise in Section 3.1, which is indeed a very common scheme in the literature, we clarify that our analysis in fact did not rely on this assumption. Towards providing you insight into the non-uniform case, we have repeated the experiments on MNIST that produced Table 1, but now with non-uniform label noise. More precisely, after uniformly randomly selecting $C$ percent of the training pairs, we proceed to contaminate the label $y_i$ in each pair $(x_i, y_i)$ in the following non-uniform manner, as outlined below in the transition kernel matrix of (True Label, Contaminated Label) entries. For example, as the matrix below indicates, if the true label $y_i = 5$, then instead of uniformly randomly drawing an alternative digit $\tilde{y}_i$ from among $\{0, 1, \ldots, 9\} \setminus \{5\}$, we have
> $\tilde{y}_i =$ \
> 0 w.p. 0.051\
> 1 w.p. 0.017 \
> 2 w.p.  0. \
> 3 w.p.  0.627 \
> $\ldots$
> | True \ Contaminated | 0     | 1     | 2     | 3     | 4     | 5     | 6     | 7     | 8     | 9     |
> |---------------------|-------|-------|-------|-------|-------|-------|-------|-------|-------|-------|
> | 0                   | 0     | 0.077 | 0.077 | 0.154 | 0     | 0.077 | 0.385 | 0     | 0.154 | 0.077 |
> | 1                   | 0     | 0     | 0.333 | 0.111 | 0     | 0.111 | 0.111 | 0     | 0.333 | 0     |
> | 2                   | 0.097 | 0.065 | 0     | 0.258 | 0.032 | 0     | 0.097 | 0.194 | 0.258 | 0     |
> | 3                   | 0     | 0     | 0.125 | 0     | 0     | 0.125 | 0     | 0.125 | 0.625 | 0     |
> | 4                   | 0.111 | 0.037 | 0.074 | 0.074 | 0     | 0.074 | 0.222 | 0.037 | 0.111 | 0.259 |
> | 5                   | 0.051 | 0.017 | 0     | 0.627 | 0.017 | 0     | 0.153 | 0     | 0.102 | 0.034 |
> | 6                   | 0.235 | 0.176 | 0.059 | 0.059 | 0.059 | 0.176 | 0     | 0     | 0.235 | 0     |
> | 7                   | 0.050 | 0.225 | 0.200 | 0.125 | 0     | 0     | 0     | 0     | 0.200 | 0.200 |
> | 8                   | 0.107 | 0.036 | 0.107 | 0.357 | 0.107 | 0.071 | 0.071 | 0.107 | 0     | 0.036 |
> | 9                   | 0.064 | 0.170 | 0     | 0.213 | 0.170 | 0.170 | 0.021 | 0.085 | 0.106 | 0     |
>
> These entries were generated by the confusion matrix of an imperfect MNIST classifier.
>
> The results from this new experiment confirm the performance benefits that were observed (compare to Table 1) under conditions of uniform label contamination.
>
> | $\epsilon_{test}$\C | 0% AT | 0% A-RRM | 5% AT | 5% A-RRM | 10% AT | 10% A-RRM | 20% AT | 20% A-RRM | 30% AT | 30% A-RRM |
> |-|-|-|-|-|-|-|-|-|-|-|
> | 0 | 96.5 | **97.3** | 93.6 | **95.6** | 60.4 | **87.8** | 32.2 | **92.4** | 58.3 | **89.2** |
> | 0.1  | 93.4 | **95.2** | 89.3 | **92.4** | 63.2 | **84.7** | 42.5 | **89.5** | 56.1 | **81.7** |
> | 0.25 | 92.4 | **93.1** | 87.9 | **90.6** | **86.9** | 86.3 | 80.6 | **89.3** | 69.0 | **79.8** |
> | 0.5 | **92.0** | 90.9 | **90.3** | 89.8 | **94.4** | 91.2 | **92.9** | 89.2 | **85.2** | 81.6 |
> | 1 | **89.4** | 85.5 | **90.3** | 86.8 | **94.9** | 92.6 | **93.9** | 86.6 | **81.8** | 77.8 |
>
> # Question
> How does RR perform relative to other state-of-the-art methods like Generalized Cross Entropy (GCE) and Early-Learning Regularization (ELR) in terms of handling noisy labels?
> # Response
> Although ELR also strives to handle noisy labels, we emphasize that RRM is a wrapper that can be paired with loss-optimization methodologies like ELR. We perform experiments on MNIST-3 (digits 0,1, and 2 only) to illustrate the effectiveness RRM wrapping. Test accuracy results are posted below.
> |  |  |  |  |
> |-|-|-|-|
> | **Method** \ **Contamination Level** | 0.55 | 0.60 | 0.65 |
> | ERM | 0.899 | 0.769 | 0.455 |
> | RRM(ERM) | 0.977 | 0.960 | **0.771** |
> | ELR | **0.987** | 0.978 | 0.545 |
> | RRM(ELR) | 0.973 | **0.984** | 0.433 |
>
> As the table shows, RRM wrapped around ERM (empirical risk minimization) improves test accuracy across contamination levels. Similarly, ELR wrapped with RRM improves test accuracy over ELR-alone, for a label contamination level of .6. While this does not appear to hold for levels 0.55 and 0.65, it is possible that  given more time, a more thorough hyperparameter search may yield similar benefits.
>
> As for GCE, this is also effectively a loss-reweighting method of sorts like RRM. Indeed, in equation (13) of GCE[1], each training example is given a weight of 1 or 0 - a byproduct of their choice of truncated, negative Box-Cox loss to be optimized. We note that RRM can reweight with more nuance however - in particular, training examples can be given weights between 0 and 1.
>
> # Question
>  How sensitive is the performance of RR to the choice of hyperparameters...?
> # Response
> Qualitatively, we find the hyper-parameters search non-burdensome, which yields beneficial RRM performance. In the discussion following Theorem 3.1, we provide insight into how $\gamma$ may be tuned in relation to an estimate $\alpha$ of the labeling error in the dataset. If invited we can quantify more precisely the sensitivity in performance changes due to hyper-parameter changes.

---

> > ### Comment · Reviewer_UZpK · 2024-08-14
> >
> > Thank you to the authors for addressing my questions. I have decided to maintain my score, which leans toward acceptance.
> >
> > I suggest that in the revised version, the authors include additional experiments beyond MNIST, such as CIFAR, CIFAR-N, and Clothing-1M, to further support the efficacy of the proposed method.

---

### Author Rebuttal · Authors · 2024-08-07

We thank the reviewers for their time and input! We have provided a separate rebuttal to each of you, and hope that we have addressed all questions/concerns. Looking forward to engaging with you in the discussion period to come.

---

### Decision · Program_Chairs · 2024-09-25

**Decision:**

Reject

**Comment:**

This paper studies learning from noisy labels. The idea here is to learn the weights on the datapoints (i.e., a loss reweighting method) to handle corrupted data. The authors draw some connections between this approach and existing frameworks. They do a set of experiments showing that their technique holds up better to large proportions of noisy labels compared to adversarial training and ERM.

The paper’s main idea is reasonable, but there’s a number of improvements that are needed before the paper reaches the bar. The reviewers asked a bunch of reasonable questions that should be addressed, especially around computational complexity. Similarly, the experiments are a bit limited (and there’s really only a very small number of baselines). For example, loss reweighting dates back to Natarajan’s 2013 paper and perhaps even earlier, and follow-up work for that technique shows how to do reweighting in a very simple way with minimal or no hyperparameter sweeps. How do  these various techniques compare? One argument the authors make is that their method can act as a wrapper and therefore be used in concert with these other noisy label techniques. I can believe this, but it is good to include the evidence as part of the next version of the paper.